# Patients’ Perceptions and Outcome Measures after Undergoing the Enhanced Transtheoretical Model Intervention (ETMI) for Chronic Low Back Pain: A Mixed-Method Study

**DOI:** 10.3390/ijerph19106106

**Published:** 2022-05-17

**Authors:** Ron Feldman, Yaniv Nudelman, Sharon Haleva-Amir, Tamar Pincus, Noa Ben Ami

**Affiliations:** 1Department of Physiotherapy, Ariel University, Ariel 4066414, Israel; nudely87@gmail.com (Y.N.); noaba@ariel.ac.il (N.B.A.); 2Department of Physiotherapy, Central District, “Maccabi” Healthcare Services, Tel-Aviv 5268104, Israel; 3School of Communication, Bar Ilan University, Ramat-Gan 5290002, Israel; sharoni.haleva.amir@gmail.com; 4Department of Psychology, Royal Holloway University of London, London WC1E 7HU, UK; t.pincus@rhul.ac.uk

**Keywords:** self-management, chronic low back pain, physiotherapy, patients’ perceptions, the Enhanced Transtheoretical Model Intervention (ETMI)

## Abstract

This study aimed to evaluate the outcome measures and perceptions of patients with chronic low back pain (CLBP) after being treated with the Enhanced Transtheoretical Model Intervention (ETMI). In this process evaluation mixed-methods study, 30 patients with CLBP electronically completed self-reported measures (function, pain, and fear-avoidance beliefs) before and after ETMI treatment. Subsequently, each patient participated in one-on-one, semi-structured interviews, which were audio-recorded, transcribed, coded, and analyzed thematically. Quantitative analysis showed significant improvements in function (*p* < 0.001), pain (*p* < 0.001), and fear-avoidance beliefs (*p* < 0.001) after receiving ETMI treatment, with a large effect size (Cohen’s d = 1.234). Moreover, the average number of physiotherapy sessions was 2.6 ± 0.6 for the ETMI intervention, while the annual average number in Maccabi is estimated at 4.1 ± 1.5. Three main themes emerged from the thematic analysis: (1) communication between the patient and the practitioner; (2) psychosocial treatment elements, and (3) ETMI as a long-term solution for CLBP. The findings of the current study highlight patients’ perceived need for an open and sincere dialogue and for receiving reassurance and encouragement about their LBP. Notably, they had no problem with the fact that they did not receive passive treatment. Accordingly, together with the significant improvement in post-treatment outcome measures, patients perceived the ETMI method as a practical tool for self-managing their back problems in the long term.

## 1. Introduction

Low back pain (LBP) is the leading cause of disability worldwide and a significant public health problem, causing a global economic burden [1]. The years lived with disability were 42.5 million worldwide in 1990, increasing by 52.7% to 64.9 million in 2017 [1]. LBP is a chronic condition with a variable course characterized by often recurrent but transient episodes of low back pain. Most episodes of LBP are short-lasting with little or no consequence, but recurrent episodes are common, and LBP is increasingly understood as a long-lasting condition with a variable course rather than episodes of unrelated occurrences [2]. Approximately 90% of LBP cases are non-specific, meaning that there is no identifiable pathoanatomical cause for the pain [3]. The current clinical guidelines for the management of non-specific LBP highlight reassurance and a return to normal activities, encouraging physical activity and the acknowledgment of psychosocial factors, as parts of a total intervention plan [3,4]. However, there is a discrepancy between evidence and practice, and the recommendations are not well implemented [5]. Research has shown that primary care clinicians still have a tendency to prescribe medication extensively, refer their patients to imaging and unnecessary procedures and consultation, and recommend rest and limitations of activity levels [5,6,7]. Some of the barriers to clinical guidelines’ recommendations are patients’ expectations of receiving a specific diagnosis and medical treatment, clinicians’ difficulty in changing their treatment habits, and a gap between clinicians’ and patients’ treatment beliefs and perceptions [5,7].

To help clinicians overcome those barriers, the Enhanced Transtheoretical Model Intervention (ETMI) was developed, designed specifically to decrease disability among patients with non-specific chronic LBP (CLBP) and to motivate them to increase their engagement in recreational physical activity [8]. ETMI is based on a behavior change theory, with the Transtheoretical Model (TTM) as its primary guiding theoretical framework. The basic premise of the TTM is to match the intervention to patients’ cognitive readiness to change by increasing their self-efficacy [8].

ETMI follows current clinical guidelines and is based on behavior change principles designed to target obstacles to physical activity by addressing self-efficacy and fear avoidance. ETMI consists of a physical examination and a discussion about the role of physical activity, which is matched to the patient’s stage of change and is guided through motivational interviewing techniques, exposure to fast walking, and goal setting. The patient receives a postcard outlining the main messages about physical activity, together with four simple stretches [8]. The intervention was tested in a pragmatic, controlled, clinical trial and was found to be both effective and cost-effective in managing CLBP [8,9].

Recent studies address issues around patients’ and physiotherapists (PTs’) perceptions and expectations of current physiotherapy and of the ETMI method [9,10]. While the patient’s primary purpose is to reduce their pain, they also expect to receive passive treatment, and information and knowledge regarding their back problem [9]. PTs have indicated that there are barriers to overcome, such as the PTs’ communication and education, an individual’s reluctance to change their daily routine, interprofessional collaboration, and healthcare complexity [10,11].

The use of mixed-methods process evaluation studies in rehabilitation research can enhance the understanding of how psychosocial factors influence LBP [12,13,14]. A critical component of this study is the interpretation of the relationships between quantitative and qualitative findings [12,13], extending the understanding beyond previous studies that use a single approach [15,16]. For example, it may provide insight into why certain interventions succeed or fail to lead to desired and effective changes in healthcare practice and patient care.

While a recent study shows that patients are beginning to perceive ETMI as an evidenced-based model for the treatment of CLBP [10], there is still a gap between patients’ perceptions of a new method that they just heard about and a method that they receive in real life. Therefore, the goals of the current study were (1) to evaluate the outcome measures of patients with LBP (pain, fear avoidance, and function) and (2) to assess the perceptions of the interventions of patients with LBP, both after receiving physiotherapy based on the ETMI method.

## 2. Methods

### 2.1. Sample Recruitment and Enrollment

A total of 30 patients with LBP admitted to a physical therapy consultation in one of Maccabi Healthcare Service (MHS)’s central physiotherapy clinics (Ramat-Gan) in the central district of Israel were enrolled in this study. Recruitment for the study was conducted by the clinics’ administration staff. Inclusion criteria were male and female patients aged 18–55 who were suffering from chronic low back pain (for over 3 months), speak and understand Hebrew, could provide written informed consent, and agreed to take part in a face-to-face interview. Exclusion criteria were dementia or severe cognitive impairments; illnesses or diagnoses that could prevent patients from participating in physiotherapy treatment; patients suffering from neurological conditions or cancer, or those who went through motor vehicle accidents; and pregnant women. The study’s participants were selected using a convenience sampling strategy, relying on age and gender to ensure a representation of varied perspectives [17]. Informed consent was obtained from all participants prior to the data collection procedures. The study was approved by the Maccabi Healthcare Service (MHS)’s IRB (Approval No. MHS-0095-20).

This mixed-methods study was performed as a preliminary step to conducting a larger prospective cohort study, aimed at investigating the ETMI method’s implementation by the Israeli healthcare services. Qualitative and quantitative data were collected concurrently, and the findings were interpreted in light of identifying inter-relations within the data [13]. The data collection and analysis procedures adopted the recommended guidelines for mixed-methods process evaluation research [12,13,18].

### 2.2. Quantitative Outcome Measures

The quantitative outcomes included were self-reported function, pain, and fear-avoidance belief ratings. Measurements were performed during patients’ admission and discharge using a customized version of Focus on Patient Inquiry Software for Therapeutic Outcomes (FOTO), Inc., Knoxville, TN, USA [19,20]. FOTO provides risk-adjusted and predictive functional scores using the Lumbar Computer Adaptive Test (LCAT) [21]. FOTO is routinely used within the MHS’s physical therapy department in all clinics.

LCAT scores range on a 0–100 linear scale, with a higher score representing a higher function [20]. The minimal clinical important difference (MCID) for the LCAT is 3–9 points, depending on the score obtained at admission. LCAT was found to be valid and accurate when compared with the Oswestry Low Back Pain Disability Questionnaire, and the English version showed high-level reliability (α = 0.92) [22,23].

The pain was rated with the Numeric Pain Rating Scale (NPRS). In the scale, the patient is asked to rate their pain intensity during the last 24 h on a scale of 0–10 (where 10 = the severest pain). This tool is widely used in the literature and is valid if compared to other pain rating scales. The MCID is 2 points [24].

Fear avoidance was measured using a modified version of the Fear-Avoidance Belief Questionnaire (FABQ) [25,26]. The modified FABQ version consists of three items, with scoring ranging from 0 to 100 and higher scores indicating higher fear-avoidance beliefs. Additionally, scores below 44 are labeled as low fear-avoidance beliefs, and scores 44 and above are labeled as high fear-avoidance beliefs [26].

### 2.3. Qualitative Phase

As for the qualitative part, a single PT, who was personally qualified, trained, and observed and supervised in the clinical filed by the ETMI’s developer (NBA), was assigned to conduct semi-structured interviews and collect field notes on nuances for the data collection in the physiotherapy clinic. A semi-structured interview guide (see Table 1) was iteratively developed by the research team based on prior experience and research evidence on the perceptions of physiotherapy treatment of patients with LBP [27,28,29].

The semi-structured interview guide was used to elicit participants’ detailed descriptions of their views and experiences of the ETMI method. Interviews were audio-recorded, transcribed verbatim, and coded. Semi-structured interviews and quantitative data collection were conducted until thematic saturation was achieved, the point at which no additional data collection would add to analysis [30].

### 2.4. Study Procedure

The ETMI method (detailed elsewhere) [8] was used by a single PT, who was qualified personally by the ETMI’s developer (NBA) and treated back pain in patients using the ETMI method for more than one year. The details of the study procedure are summarized in Table 2.

### 2.5. Data Analysis

#### 2.5.1. Sample Size

The sample size was calculated with G*Power 3.1.9.4 for a paired *t*-test to detect the difference between two dependent means. The input parameters were as follows: for a single-tailed test, assuming a medium effect size of 0.5, α = 0.05, and power = 0.8, the total sample size recommended was 27 participants.

#### 2.5.2. Quantitative Data

Quantitative data analysis was performed using IBM SPSS Statistics version 27 (IBM Corp., Armonk, NY, USA). Sociodemographic and outcome scores were described using frequencies, means, and standard deviations (SDs). The Shapiro–Wilk test was used in order to test for normality of the data. In order to explore the differences before and after the intervention, we used a paired t-test for normally distributed variables and Wilcoxon signed-rank test for non-normally distributed variables. Correspondingly, effect sizes were calculated using Cohen’s d and Pearson’s r [31]. Additionally, we used McNemar’s chi-squared test to assess differences in the proportions of dichotomized outcomes (FABQ).

#### 2.5.3. Qualitative Data

The qualitative analysis was based on a thematic analysis of the interviews. Following Brown and Clarke’s [32] six-step process, we thematically analyzed the interview transcripts to identify the main issues that rose from the text. To become familiar with the data, three of the authors (R.F., Y.N., and S.H.A.) independently identified prevalent themes within the texts. Upon the completion of the individual identification of the prominent themes, these above-mentioned authors reviewed them to reach an agreement on three leading themes and to define and name them. Finally, all interviews were re-coded by the authors according to the new thematic scheme.

#### 2.5.4. Mixed-Methods Process Evaluation Analyses

Mixed-methods analysis began by integrating individual participants’ quantitative measures and exemplar quotes into a joint display. A joint display is a recommended tool for mixed methods in order to facilitate the interpretation and understanding of relationships among data [33]. Next, prevalent patterns of outcome measure variables, individual function, pain, and fear avoidance were identified within the quantitative data. Finally, an iterative process was undertaken to interpret and describe how the participants experienced the ETMI method, while identifying the significant themes emerging from the interviews. The themes’ exemplary quotations were combined with the quantitative findings.

## 3. Results

### 3.1. Participant Characteristics

All the 30 patients enrolled in this study completed the intervention. A description of the patients’ characteristics are presented in Table 3.

#### Treatment Session

An average of 2.6 ± 0.6 physiotherapy treatment sessions were administered during the ETMI intervention (Table 3). Comparatively, MHS’s average annual number of physiotherapy sessions for the treatment of LBP is 4.1 ± 1.5.

### 3.2. Quantitative Analysis: Patients Report Outcome Measures

A summary of the quantitative analysis is presented in Table 4.

#### 3.2.1. Function

The Shapiro–Wilk test indicated that all variables representing function were normally distributed. The paired sample t-test revealed a positive significant change (t(29)= 11.42, *p* < 0.001) between the functional score measured at admission (M = 46.27, SD = 12.35) and the functional score measured at discharge (M = 71.93, SD = 9.47), with a large effect size [34]—Cohen’s d = 2.085. Furthermore, the functional score measured at discharge significantly differed from the LCAT’s risk-adjusted predicted post-treatment functional score generated at admission (M = 59.37, SD = 8.85); t(29)= 6.75, *p* < 0.001, with a large effect size—Cohen’s d = 1.234.

#### 3.2.2. Pain Rating

The Shapiro–Wilk test indicated that pain ratings at admission were normally distributed; however, the normality assumption was denied for pain levels at discharge. A Wilcoxon signed-ranks test indicated that pain at discharge (median = 6) was lower than pain at admission (median = 1.5); z = −4.8, *p* < 0.001.

#### 3.2.3. Fear-Avoidance Beliefs (FABs)

The exact McNamar’s test revealed a statistically significant reduction in fear-avoidance beliefs post-intervention compared to pre-intervention (χ2 = 15.05 *p* < 0.001).

### 3.3. Qualitative Analysis: Themes and Sub-Themes

Three main themes emerged from the interviews, and each theme is composed of a few sub-themes (see Table 5): (1) communication between the patient and the practitioner (three sub-themes); (2) psychosocial treatment elements (three sub-themes); and (3) ETMI as a long-term solution for CLBP (two sub-themes).

A coding scheme with quotations underpinning the variety of the sub-themes is provided in Table 5.

#### 3.3.1. Communication between the Patient and the Practitioner

##### Being Attentive

Most patients noted that one of the most significant things that they experienced during the first treatment was the therapist’s attentiveness to their problems. Patients noted that it was important for them to have someone who was attentively listening to their problems, without interrupting them.

##### Patient–Practitioner Dialogue

Furthermore, all patients emphasized the importance of the initial dialogue between the practitioner and the patient during the first session. They all mentioned the importance of having an open and sincere dialogue, as it enables the patient to take an active part in the process. They stated that it made them feel like they were taking part in their healing process. Most of them maintained that it was the first time they experienced an open dialogue with a practitioner.

##### In-Depth Explanation

All patients stated that they received an in-depth explanation as a result of the dialogue, which, in turn, helped them to obtain a wider perspective of their health problem.

#### 3.3.2. Psychosocial Treatment Elements

##### Being Reassured and Increasing Self-Confidence

Most patients noted that, throughout the treatments, they felt that they gained a great deal of self-confidence in handling their back problem, together with a sense of reassurance, which was given by the practitioner. Most stated that they felt much more calmer and reassured regarding their back pain as the treatments progressed.

##### Letting Go of Fear

Some patients mentioned that their sense of fear, specifically the fear of pain and causing damage to their back, diminished throughout the treatments.

##### Increasing Patient’s Self-Efficacy

Finally, most patients noted that their sense of empowerment and self-efficacy in returning to normal while performing their daily routine increased significantly during the treatments.

#### 3.3.3. ETMI as a Long-Term Solution for CLBP

##### Practical Tools for Self-Managing LBP

Most patients agreed that the treatment they underwent gave them some practical tools to self-manage their back problem in the future. Among other things, most patients maintained that they adapted to a healthier lifestyle, with an emphasis on regular and daily exercise, in order to avoid, as much as possible, future back pain. In addition, they noted that the tools they received during the treatments provided them with the information they were missing and that, currently, they are able to self-manage their back problem in a much better way than before.

##### Patients’ Insights from The Treatment

All patients gave their insights into the treatment they experienced. All the insights were positive and referred to their better understanding of their LBPdue to the treatment. Patients indicated their understanding that they could return to their daily function and activities, even though their back pain was not completely resolved.

## 4. Discussion

This study focused on the perceptions and outcome measures of patients with CLBP following physiotherapy treatment using the ETMI method. By using mix-methods research, this study obtained a more comprehensive view of patients’ treatment perceptions, relationships with clinicians, and lived experience of their problems following physiotherapy treatment using the ETMI method [35,36].

Patients who were treated with the ETMI method not only demonstrated significant improvement in their outcome measures but also perceived ETMI as a positive and applicable method to treat their back problem.

The qualitative analysis indicated that patients perceived the ETMI method as a practical tool for self-managing their back problems for the long term. Specifically, patients’ perceived the need for an open and sincere patient–practitioner dialogue and for receiving reassurance and encouragement about their back pain. Patients also indicated that treatment using the ETMI method helped them regain their self-confidence and self-efficacy to overcome their pain.

Notably, patients did not have any problem not receiving passive treatment as part of their physiotherapy session.

Our findings are consistent with those of studies that indicate that patients seek a confidence-based relationship with their caregivers [10,11,37,38]. Moreover, the findings regarding patients’ perceptions of the ETMI method as a practical tool for the treatment of back pain are consistent with those of several studies addressing patients’ perceptions of LBP clinical guidelines [15,39].

The quantitative analysis demonstrated significant changes in all outcome measures (*p* < 0.001), with large effect sizes where applicable. Notably, a large effect size was recorded between the post-treatment LCAT’s risk-adjusted predicted functional score and the functional score measured at discharge. These data indicated that the change in the functional score was not only significant between the admission and discharged scores but also significantly better than the improvement predicted by the FOTO logarithm. Furthermore, the quantitative analysis demonstrated a significant reduction in fear avoidance among all patients. Of the 17 patients previously categorized as having high fear avoidance, all moved to the low category. This, in fact, reinforces the results of previous studies on ETMI that have shown that ETMI is effective in reducing disability and pain and improving self-reported outcome measures, e.g., mental and physical health [8,9].

Moreover, it is interesting to note that an average of 2.6 ± 0.6 physiotherapy sessions were carried out during the ETMI intervention (Table 3). Comparatively, Maccabi’s average annual number of physiotherapy sessions for the treatment of LBP is 4.1 ± 1.5. This suggests that, for ETMI, fewer sessions are more likely to be needed. This fact also reinforces the findings of previous studies on ETMI, which have shown that ETMI is also effective in reducing the number of physiotherapy treatments compared to physiotherapy as usual (3.5(CI 3.2–3.9) vs. 5.1(CI 4.4–5.7), respectively) [8,9].

The qualitative findings can be used to provide an explanation for the improved pain, function, and fear scores measured quantitatively. Self-efficacy, self-confidence, reassurance, and fear avoidance reduction were mentioned by patients as factors that helped them to overcome their pain, fears, and functional difficulties. These findings are in line with the evidence suggesting that psychosocial factors contribute to the maintenance of CLBP [2,40,41,42]. Specifically, studies show that self-efficacy [42], patients’ education [43], patients’ reassurance [44,45], and low levels of fear avoidance [40,46] are protective factors for pain long-term development.

### 4.1. Study Limitations

The interviews were conducted by the same PT who provided the treatment according to the ETMI method. This may entail a risk for social desirability and interviewer biases [33]. However, patients were discharged, no dropout was recorded, and they did not return to another treatment session after 3 months of follow-up.

Furthermore, to reduce the risk, all the quantitative data were collected privately without the presence or assistance from the interviewer.

### 4.2. Study Implications for Physiotherapy Practice

This study’s findings may help PTs understand that patients with CLBP perceive ETMI as an applicable and manageable method that can help them to cope and self-manage their back pain. Moreover, it can be expanded to some other chronic pain conditions such as other musculoskeletal chronic pain disorders.

## 5. Conclusions

The novelty and importance of this study is that patients with CLBP had no problem receiving no passive intervention such as a massage. Patients perceived ETMI as a method that can help them by providing practical tools to self-manage their back problems, reassuring them, and improving their self-efficacy and self-confidence to cope with their pain. This study highlights the uniform and positive relationships between treatment outcome measures (disability, pain, and fear avoidance) and patient treatment perceptions.

## Figures and Tables

**Table 1 ijerph-19-06106-t001:** Semi-structured interview guide.

Primary Question	Example Probe
1. Tell me how did you feel throughout the treatments?	Can you tell me how did you feel throughout the initial assessment? Can you tell me how did you feel throughout the first/second treatment? Let us talk about the first treatment: what did you like referring to it? What did you dislike in reference to the treatments? If there is one thing you took out of this treatment, what was it? Had the treatment met your expectation?
2. Tell me how do you manage your back problem?	What are you doing with your back pain? How did the treatment affect the way you manage your back problem? What do you do differently today (referring to your back problem)?

**Table 2 ijerph-19-06106-t002:** Study procedure based on the four parts of the ETMI method.

Part	Details
Pre-intervention	Signing a personal consent form Answering an opening session FOTO questionnaire on
1. Creating a therapeutic alliance	communication skills and reassurance
2. Clear messages to the patient—three mandatory sentences	“Physical activity is the only thing that will help your back pain over time” “It’s easy to reduce your pain now—but the important thing is to prevent the next episode and help you to self-manage your back pain” “Your body must be strong and flexible”
3. Exposure to brisk walking and graded activity	Brisk walking in the corridor, hand in hand with the therapist
4. Postcard, booklet, infographics, and short videos	Postcard with reminder messages on how to self-manage LBP, and infographics, a booklet, and two short animated videoclips on facts and myths about low back pain
Post-intervention	Answering a closing session FOTO questionnaire A face-to-face, one-on-one interview

FOTO—Focus on Therapeutic Outcomes.

**Table 3 ijerph-19-06106-t003:** Descriptive characteristics of the patients (*n* = 30).

Variable	Participants (*n* = 30)
Gender *n* (%)	
Female Male	14 (46.6%) 16 (53.3%)
Average Age Average BMI	37.9 ± 11.09 27.6 ± 5.3
Education Level (*n*%)	
Secondary	3 (10%)
Post-secondary	9 (30%)
Academic	18 (60%)
Employment status *n* (%)	
Employed	24 (80%)
Unemployed	6 (20%)
Duration of symptoms (>3 months) (*n*%)	30 (100%)
General health status (*n*%)	
Healthy	30 (100%)
Physiotherapy treatment for LBP in the past (*n*%)	
Yes	18 (60%)
No	12 (40%)
Referring factor for Physiotherapy (*n*%)	
GP	5 (16.66%)
Orthopedic	19 (63.33%)
Self-referral	6 (20%)
Orthopedic surgeries for LBP in the past (*n*%)	
No	30 (100%)
Medication treatment for LBP (*n*%)	
None	12 (40%)
Pain relief (e.g., paracetamol)	9 (30%)
NSAIDs (e.g., ibuprofen)	6 (20%)
Opioids (e.g., oxycodone)	3 (10%)
Physical Activity sessions per week (*n*%)	
None	6 (20%)
1–2 sessions	11 (36.66)
2–3 sessions	6 (20%)
3–4 sessions	7 (23.33%)
Average number of physiotherapy sessions	2.6 ± 0.6

Frequencies and descriptive statistics of study participants. All variables are presented as frequencies (*n*%), except for the ‘average age’ and ‘average number of physiotherapy sessions’ variables, which are presented as mean ± standard deviations. BMI—Body Mass Index; LBP—low back pain; GP—General Practitioner; NSAIDs—Non-Steroidal Anti-Inflammatory Drugs.

**Table 4 ijerph-19-06106-t004:** Summary of the quantitative analysis (*n* = 30).

		Admission	Predicted	Discharge	Change		Statistic		Effect Size	
					Discharge/Admission	Discharge-Predicted	Discharge/Admission	Discharge-Predicted	Discharge/Admission	Discharge-Predicted
Functional status Mean ± SD		46.27 ± 12.35	59.37 ± 8.85	71.93 ± 9.47	25.66 ± 12.3	12.56 ± 10.18	11.42 *	6.75 *	d = 2.085	d = 1.234
Pain	Median IQR	6 4.75–8	N/A	1.5 1–2.25	4 3–5.25		z = −4.8 *		r = −0.62	
FAB Frequency	High	17 13	N/A	0 30	−17 +17		χ2 = 15.05 *		N/A	
	Low									

* *p* < 0.001; IQR—inter-quartile range; SD—standard deviation; FAB—fear-avoidance beliefs; N/A—not applicable; d—Cohen’s d; r—Pearson’s r; FAB high—measured as participants with FAB score > 44; FAB low—measured as participants with FAB score < 44.

**Table 5 ijerph-19-06106-t005:** Theme and sub-theme scheme.

Main Themes	Sub-Themes	Quotations
Communication between the patient and the practitioner	Being attentive	P2—“Throughout the treatments, I felt that someone is really listening to me, that someone gives me a place to express myself, that someone is really taking care of my back problem”.
		P8—“…. I liked the fact that you have let me express my pain and especially that you were referring to every note I have told you and didn’t ignore my feelings”.
		P14—“You listened to every word that came out of my mouth. You gave me a place to express myself. I really appreciate it”
		P17—“…You have listened to everything that came out of my mouth. This is the first time it has happened to me; a practitioner who listens to me like this.”
	Patient–practitioner dialogue	P6—“There is no doubt that our open dialogue made me understand some important things about my back”.
		P12—“I think the most essential thing in the whole treatment was the open conversation we had together. I told you how I feel and you gave me a place to express myself. I felt like I was really part of the process itself. I do not think I have so far been able to be in such treatment”
		P25—“To tell you the truth, this is the first time I have experienced such an open dialogue with a practitioner. Usually, after 5 min they already tell me what my problem is and send me home.... but this time I really feel that someone wants what’s best for me and even gives me an opportunity to make some decisions regarding my back problem”.
	In-depth explanation	P11—“I liked your explanations. They gave me a more up-to-date and accurate perspective on my back problem”.
		P18—“You explained in great detail why my back hurts and how low back pain is treated according to global guidelines. You explained things to me that I did not know and now I understand them much more deeply”.
		P21—“All the explanations you gave me about back pain were very clear and very detailed. I loved that I get the most up-to-date and detailed information”.
Psychosocial treatment elements	Being reassured and increasing self-confidence	P1—“Throughout the treatments you gave me reinforcements that reassured me greatly…. I feel I have much more confidence since I have met you”.
		P17—“The treatments gave me self-confidence to go back to do things that I used to do previously”.
		P28—“The main thing I remember from the treatments is your reassurance. You gave me all kinds of messages that had reassured me, and I think that is what has helped me regaining self—confidence to do things again”.
		P29—“The treatments were conducted in a very relaxed atmosphere. I think it really calmed me down and my troublesome thoughts about my back problem. I really felt like during the treatments I became much more relaxed”.
	Letting go of fear	P5—“I was afraid. I was afraid to do more damage, I was afraid the pain would get worse. I was just afraid to move. But you taught me that fear does not advance me at all in the process and that I slowly have to get rid of it. I started moving my body. The fear was there but much less”.
		P9—“you showed me all sorts of movements that I could perform and that I should not be afraid of”.
		P11—” …I have realized I am capable of doing things I was afraid to do”.
	Increasing patient’s self-efficacy	P3—“Suddenly my sense of self-efficacy returned to what it was. I realized that I could and should do things that were difficult for me in the past and that I would not hurt my back”
		P20—“The treatments had strengthened my sense of self-efficacy. I suddenly realized I could do things I could not have done before”.
		P28—“During the process, you told me I have to get back to my normal activities. This advice had helped me to return to my daily routine. I had realized that my back pain does not bother me terribly.”
ETMI as a long-term solution for CLBP	Practical tools for self-managing LBP	P17—“well….If my back will hurt… I know I will rest a bit and probably the next day I will go for a brisk walk or for a run”.
		P20—“Now I know that practically I need to rest when it hurts and when it does not hurt start moving the body”
		P22—“Now I know exactly what I have to do the next time my back will hurt”.
	Patients’ insights from the treatment	P3—“I understood that I can totally self-manage my back problem. I realized that it was mostly up to me as low back pain is a part of our daily routine and that I can completely manage it on my own”.
		P7—“I understood that if I will experience back pain again, I will know how to handle it. I will probably rest a bit, do some relaxing exercises, walk a bit outside and will not fear it”.
		P14—“I understood that I do not have any serious back problem and that I can perform regular physical activity on a daily basis, even if it is only for 10 min a day.”
		P23—“I understood that I have to work hard and to strengthen my whole body because that is what will eventually make my back stronger and relive the pain. Nowadays I mostly continue to do my daily routine exercises—otherwise, it will not work.”

## Data Availability

Not applicable.

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
