# Peer review of "Patients’ Perceptions and Outcome Measures after Undergoing the Enhanced Transtheoretical Model Intervention (ETMI) for Chronic Low Back Pain: A Mixed-Method Study"

_ijerph, 2022, doi:10.3390/ijerph19106106_

Round 1

Author Response

Reviewer 1:

I am very interested in the subject of the author's research, which is undoubtedly good for this group, but the research still has certain problems in some aspects, and I hope the author can improve it.

Answer: We would like to thank the reviewer for finding our manuscript interesting. We devoted considerable attention to the reviewer's comments and revised the manuscript accordingly.

  • The authors of the abstract should add a description of the innovation of the article. This study is different from previous studies and highlights the innovation of this study;

Answer: We thank the reviewer for the important remark. We changed the sentence accordingly.

The findings of the current study highlight patients' perceived need for an open and sincere dialog and for receiving reassurance and encouragement about their LBP. Of importance, they had no problem with the fact that they did not receive passive treatment. Accordingly, together with significant improvement in post-treatment outcome measures, patients perceived the ETMI method as a practical tool for self-managing their back problems in the long term.  (page 1, line 24)

  • In the introduction section on the first page, the author does not state what the background of the study is, especially the lack of an introduction to the international background;

Answer: We thank the reviewer for the important remark. We added some introduction to the international background.

Years lived with disability were 42.5 million worldwide in 1990, increasing by 52.7% to 64.9 in 2017 [1]. LBP is a chronic condition with a variable course characterized by often recurrent but transient episodes of low back pain. Most episodes of low back pain are short-lasting with little or no consequence, but recurrent episodes are common and low back pain is increasingly understood as a long-lasting condition with a variable course rather than episodes of unrelated occurrences.[2] Approximately 90% of LBP cases are non-specific, meaning that there is no identifiable pathoanatomical cause for the pain [3]  (Page 1, Lines 49-56)

  • On the second page, the author lacks a theoretical analysis of the article, which makes the article lack certain theoretical support;

Answer: We added theoretical analysis support

ETMI based on a behavior-change theory (Transtheoretical Model [TTM]) as its primary guiding theoretical framework. The basic premise of the TTM is to match the intervention to patients’ cognitive readiness to change, by increasing their self-efficacy [8]. (Page 2, lines 70-73)

  • In the method introduction section, the author did not clarify the source of the data, especially the details of the data and data, and asked the author to explain the details of the data source;

Answer: We have deleted the method introduction section and changed the overall method section accordingly.

  • In 2.2. In the quantitative approach, I hope the author can be more specific, and I can't see the details yet;

Answer: Quantitative approach was changed to 'quantitative outcome measures'. (page 4, line 161). Details on the outcome measures are presented in the same paragraph as well in the result quantitative section.

  • In 2.5.1., the author should give a detailed introduction to the software, including the software company, city and other information.

Answer: details were added (IBM Corp., Armonk, NY, USA). (Page 4, lines 216-217).  

  • In section 2.5.2., I hope the author can introduce it more clearly, it is still vague at present.

Answer: This section was re-written in accordance with the reviewer's suggestion.

The qualitative analysis was based on a thematic analysis of the interviews. Following Brown and Clarke’s [32] six step process and while referring to the interviews’ transcription, we thematically analyzed them to identify the main issues that rose from the text. To become familiar with the data, three of the authors (RF, YN and SHA) independently identified prevalent themes within the texts. Upon the completion of the individual identification of the prominent themes, these above-mentioned authors reviewed them to reach an agreement on three leading themes, define and name them. Finally, all interviews were re-coded by the authors according to the new thematic scheme. (Page 6, lines 226-283).

  • In section 3.3.1., can the author provide more structured interviews about patients, or interviews with patients, it would be better if they could increase the contrast;

Answer: We thank the reviewer for this comment. This entire section was changed and simplified. The themes, sub-themes, and quotations are now in table 5 (page 10, line 383) while the section remains with qualitative result interpretation only.  (page 8, from line 290)

  • In the Discussion section, the author lacks a dialogue with the existing research, and cannot clearly show the research findings of this paper, so some references can be added.

Answer: The discussion was re-organized and re-written according to the reviewer's suggestions. More evidence-based explanations together with some more references were added and incorporated into this section. (Page 12, from line 400).

  • In the conclusion part, the author discusses less, does not highlight the innovation of this study, and the conclusion does not explain the theoretical contribution.

Answer: the conclusion part was re-written.

The novelty and importance of this study is that CLBP patients had no problem accepting no passive intervention such as massage. Patients perceived ETMI as a method that can help them in providing practical tools to self-manage their back problems, reassure them and improve their self-efficacy and self-confidence to cope with their pain. This study highlights uniform and positive relationships between treatment outcome measures (disability, pain, and fear avoidance) and patient treatment perceptions. (Page 13, Line 456).

  • The author does not state the applicability of this study, especially whether it can be extended to other groups or regions;

Answer: the applicability of this study was elleborated in 4.2. Study Implications for Physiotherapy Practice.

Moreover, it can be expended to some other chronic pain conditions such as other musculoskeletal chronic pain disorders.  (Page 13, Lines 453)

  • Hope that the author can explain the research method used in this article, and explain why this research method was chosen and not other research methods;

Answer: We would like to thank the reviewer for this important comment.

We want to draw the reviewer attention to the last paragraph in the introduction:

The Use of mixed-methods process-evaluation studies in rehabilitation research can enhance understanding of how psychosocial factors influence LBP [12-14]. A critical component of this study is the interpretation of the relationships between quantitative and qualitative findings [12,13], extending understanding beyond previous research that uses a single approach [15,16]. For example, it may provide insight into why certain interventions succeed or fail to lead to desired and effective changes in health care practice and patient care. (page 3, line 115)

  • The author is expected to revise the reference format of this article to be consistent with the research published in the journal, especially to provide specific page numbers

Answer: the references section was revised and re-written.

Reviewer 2 Report

Greetings, authors!
Thank you for providing me with the opportunity to review your excellent work. I hope that my suggestions will help to improve the quality of your publications. Attached is a word document with all of the points.

Author Response

Reviewer 2:

We would like to thank the reviewer for finding our manuscript interesting. We devoted considerable attention to the reviewer's comments and revised the manuscript accordingly.

  • Title:

Specific topic: Title should reflect as ETMI as an intervention. As a result, patient perception and CLBP outcome have improved.

Answer: We would like to thank the reviewer. We changed the title according to the reviewer suggestion

Patients' perceptions and outcome measures after undergoing the Enhanced Transtheoretical Model Intervention (ETMI) for chronic low back pain: a mixed-method study

  • Novelty & originality: Previous studies by the same authors looked at pretreatment patient and PT perceptions of ETMI and were published in 2020 and 2021, and this is a continuation of those studies.

Answer: We would like to thank the reviewer for this comment. We added a paragraph in the introduction regarding our previous studies. (Page 2, Line 82-88).

Recent studies address issues around patients' and physiotherapists (PT's) perceptions and expectations of current physiotherapy and of the ETMI method [9,10]. While patient's primary purpose is to reduce their pain, they expect to receive also passive treatment, and information and knowledge regarding their back problem, [9] PT's indicated that there are barriers to overcome, such as PTs communication and education, individual's reluctance to change their daily routine, interprofessional collaboration and health care complexity [10].

  • Abstract: Add the number and duration of ETMI intervention sessions, as well as whether there is any other control intervention, and more clarity to quantitative and qualitative outcome measures and their interpretation, lines 17-22, which should be elaborated in the text's result section. Add as reduced treatment sessions in the result is another addition to ETMI, as compared to control, adds another clinical significance.

Answer: We change the abstract following the reviewer comment.

Abstract: The study aimed to evaluate chronic low back pain (CLBP) patients' outcome measures and perceptions after being treated with the Enhanced Transtheoretical Model Intervention (ETMI). In a process evaluation mix method study, 30 CLBP patients completed electronically self-reported measures (function, pain, and fear-avoidance belief) before and after an ETMI treatment. Subsequently, each of the patients participated in a one-on-one semi-structured interview, which were audio-recorded, transcribed, coded, and analyzed thematically. Quantitative analysis showed significant improvements in function [P<0.001], pain [P<0.001], and fear-avoidance beliefs [P<0.001] after receiving ETMI treatment. With a large effect size [Cohen's d=1.234]. Moreover, the average number of physiotherapy sessions was 2.6±0.6 for the ETMI intervention while the annual average number in Maccabi is estimated at 4.1±1.5. Three main themes emerged from the thematic analysis: 1) Communication between the patient and the practitioner; 2) Psycho-Social treatment elements and 3) ETMI as a long-term solution for CLBP. The findings of the current study highlight patients' perceived need for an open and sincere dialog and for receiving reassurance and encouragement about their LBP. Of importance, they had no problem with the fact that they did not receive passive treatment. Accordingly, together with significant improvement in post-treatment outcome measures, patients perceived the ETMI method as a practical tool for self-managing their back problems in the long term.

We have also Added a subheading 'treatment sessions' in the result section. (page 7, line 255).

3.1.1. Treatment session

An average of 2.6±0.6 physiotherapy treatment sessions were administered during the ETMI intervention (Table 3). Comparatively, Maccabi's average annual number of physiotherapy sessions for treating LBP is 4.1±1.5.

  • Introduction:

Paragraph-1: In the introduction, there should be a link between why the authors started observing LBP and why they ended up studying CLBP.

Answer: we added a sentence in the introduction section that links between LBP to non-specific CLBP.

LBP is a chronic condition with a variable course characterized by often recurrent but transient episodes of low back pain. Most episodes of low back pain are short-lasting with little or no consequence, but recurrent episodes are common and low back pain is increasingly understood as a long-lasting condition with a variable course rather than episodes of unrelated occurrences.[2] Approximately 90% of LBP cases are non-specific, meaning that there is no identifiable pathoanatomical cause for the pain [3]. (Page 2, Lines 50-56)

  • Paragraph 2: The difference between the Transtheoretical modal and the enhanced version, as well as its role in physical therapy, must be explained

Answer: we added a sentence in the second paragraph clarifying the differences between ETMI to the Transtheoretical modal.

ETMI based on a behavior-change theory (Transtheoretical Model [TTM]) as its primary guiding theoretical framework. The basic premise of the TTM is to match the intervention to patients’ cognitive readiness to change, by increasing their self-efficacy [8].

(Page 2, lines 70-73)

  • Paragraph-3: Add a third paragraph as needed for this continuity study, based on the authors' previous studies on pretreatment patient and PT perceptions of ETMI, which were published in 2020 and 2021.

Answer: We added a third paragraph as needed for this continuity study, based on the authors' previous studies on pretreatment patient and PT perceptions of ETMI, which were published in 2020 and 2021.

Recent studies address issues around patients' and physiotherapists (PT's) perceptions and expectations of current physiotherapy and of the ETMI method [9,10]. While patient's primary purpose is to reduce their pain, they expect to receive also passive treatment, and information and knowledge regarding their back problem, [9] PT's indicated that there are barriers to overcome, such as PTs communication and education, individual's reluctance to change their daily routine, interprofessional collaboration and health care complexity [10].

(Page 2, lines 82-88)

  • Instead of quoting another study's methodology in such detail [Lines: 54 -78 insignificant in such detail in introduction], avoid subheadings in the introduction and conclude ETMI reliability and validity in a conclusive manner.

Answer: We have deleted and changed the paragraph accordingly.

  • Paragraph-4: Add a fourth paragraph to explain why the current study's quantitative and qualitative variables were chosen.

Answer: We elaborate the paragraph explain why the current study's quantitative and qualitative variables were chosen.

The Use of mixed-methods process-evaluation studies in rehabilitation research can enhance understanding of how psychosocial factors influence LBP [12-14]. A critical component of this study is the interpretation of the relationships between quantitative and qualitative findings [12,13], extending understanding beyond previous research that uses a single approach [15,16]. For example, it may provide insight into why certain interventions succeed or fail to lead to desired and effective changes in health care practice and patient care.

. (Page 3, Lines 115-121)

  • last paragraph: should include a problem statement, why is it important to study, line-85 be specific about the patient's perception before, during, and after the study?

Answer: The last paragraph was re-written.

While recent study shows that patients are beginning to perceive ETMI as an evidenced-based models for the treatment of CLBP, [10] there is still a gap between patients' perceptions of a new method that they just heard about to a method they receive in real life. Therefore, the goals of the current study were: (1) to evaluate LBP patient's outcome measures (pain, fear avoidance and function); and (2) to assess LBP patients’ perceptions of the intervention, both after receiving physiotherapy based on the ETMI method. (Page , lines 122-128)

English format: The hyphen between words in the overall text must be removed. Eg: words in lines: 15,15, 16, 18, 19, 20, 22, 23, 27, 33, 35, 44, 46, 52, 55, 56, 69, 70, 72, 79, 82, 87, 88, 95, 97, 104, 107, 108, 111, 113, 114, 116, 118, 128, 133, 154, 156, 163, 165, 172, 177, 178, 180, 199, 214, 216, 228, 229, 231, 238, 251, 255, 257, 303, 304, 346, 361, 373, 374

Answer: The hyphen between words in the overall text removed

  • Methods:

The reader will be confused by the overall method paragraphs, so please keep it simple and orderly.

Start with the subject's characteristics, such as lines 97-109 in the first paragraph.

Answer: We simplify the overall method sections according to your review. (Page 3, from line 128)

  • Descriptive data: age/sex/BMI [add table-3], sample size: how/sampling method/adequate with appropriate population should be mentioned in methods.

Answer:

gander: males and females, was added in page 4, line 143.

BMI: was added to table 3. (page 7)

Sample size was added to data analysis: 2.5.1. Sample size. (page 6, line 210)

The sample size was calculated with G*Power 3.1.9.4 for a paired t-test to detect the difference between two dependent means. The input parameters were as follows: for a single-tailed test, assuming a medium effect size of 0.5, α= 0.05 and power= 0.8, the total sample size recommended was 27 participants.

  • 4th paragraph & subheading: Quantitative approach: explained well

Answer: Thank you very much

  • 5th paragraph & subheading: Qualitative approach: instead of 'first authors, RF, PhD student' [lines 132, 150] and 'guide' line-154, it could be single PT assigned to conduct semi structured interview' or FOTO's interpretation of data.

Answer: Instead of 'first authors, RF, PhD student' [lines 132, 150] and 'guide' line-154, we changed it to  - a single PT, who was personally qualified, trained, observed and supervised in the clinical filed by the ETMI's developer (NBA), (page 4, line 183), a single PT (page 5, line 198) and semi structured interview (page 4, line 190)

  • Lines 87-95 should be placed in the second paragraph.

Answer: Lines 87-95 were placed in the second paragraph under the subheading 2.1. Sample recruitment and enrollment. (Page 4, line 155)

  • What is the PT's qualification or any specific ETMI training certification for each step?

Answer: details on ETMI qualification were added (page 4, Line 183, page5, line 198)

As for the qualitative part, a single PT, who was personally qualified, trained, observed and supervised in the clinical filed by the ETMI's developer (NBA), assigned to conduct semi structured interview.

  • Table-1, 2, 3, 4, 5: to be aligned properly

Answer: Table-1, 2, 3, 4, 5 were realigned

  • Lines 149-152 should be moved to the third paragraph, with the number of sessions and duration of each ETMI?

Answer: The entire method section was reorganized accordingly to the suggestions.

  • Is there any other conventional PT treatment [Line332] that CLBP patients have received? If so, please elaborate

Answer: We added a continuation for the sentence so it will be clear that ETMI was used as the single physiotherapy treatment method.

…given through the ETMI method. (Page 12, line 390)

  • As it was a CLBP subject, did you have any other red flag screening?

Answer: Red flag screening is an integral part of the physical examination in ETMI method and it's detailed in the largest ETMI trail (reference 8)

  • Lines 90 and 99 – What is the purpose of [citations 10 and 17] in this context?

Answer: citations were deleted.

  • Communication between the patient and the practitioner Is it patient-practitioner on line-235?

Answer: the main them has been changed according to the reviewer suggestion Communication between the patient and the practitioner (page 8, line 290).

  • P2, P8, P17, [Line 241-327]... It could be written better; rather than simply quoting as a communication, it should be written in the researcher's interpretation and conclusive style.

Answer: We would like to thank the reviewer for this comment. Although in a qualitative design study the results should be written as summaries of the findings with some highlights quotations from participants, we re-wrote this section in accordance with the reviewer suggestion. Furthermore table 5- main themes and subthemes was emerged and recombined with appendix A table, that is now representing all quotations together with the main themes and the subthemes.

  • Discussion: More evidence-based explanations are needed in the discussion section for each qualitative and quantitative variable influenced by ETMI.

Answer: The discussion was re-organized and re-written according to the reviewer suggestions. More evidence-based explanations were incorporated into this section. Page 12, line 400

of note is that patients didn’t have any problem not getting any passive treatment, as part of their physiotherapy session. Our findings are consistent with studies that indicated that patients seek a confidence-based relationship with their caregivers [10,37,38]. Moreover, patients' perceptions of the ETMI method as a practical tool for the treatment of back pain, is consistent with several studies, addressing patients perceptions of LBP clinical guidelines [39,40].

  • Any role of gender, educational/employment/past PT/medication/number of sessions/week in pre-intervention and post-intervention ETMI to be discussed?

Answer: Those characteristics were taken as an integral part of the data gathering and participants recruitment and none of them was discussed or elaborated in the scope of this paper.

  • Future implications of the study: Why only for CLBP, and not for other diseases

Answer: We have elaborated this paragraph in accordance with the reviewer comment.

Moreover, it can be expanded to some other chronic pain conditions such as musculoskeletal chronic pain disorders. (Page 13, line 453).

Reviewer 3 Report

Title

Title is appropriate because it is completely informative about the contents of the paper.  

Abstract

The abstract respects the rules of the journal. The background and the aim are interesting. In the design is present the type of study. Setting, population and methods need to be better explained. The clinical Impact is present but need to be better explained.

Text

The introduction of the study does clearly sum up the background of the study.

The authors provide a rationale for performing the study based on a review of the medical literature, but they can improve it. Furthermore, they do not define well terms used in the remainder of the manuscript. (Please add a list of abbreviations before References section to your manuscript).

The hypothesis is defined.

The methods are not clear about the statistical method.  

First, the authors can add a subtitle of sample size estimation right behind study population; based on what did you calculate this sample size?

The methodology needs to be improved a bit.

About the patients: how many patients are recruited? Because it is not clear. This is a crucial information.

About the centers: Which ones are they?

The results are reported clearly and concisely.

In the discussion, it is important to emphasize the importance of this method explaining the specific aspects in the daily practice.

Limitation:

To be explained better.

Tables

They sum up the study concisely and clearly.

Figures

They sum up the study concisely and clearly.

General comments

The purpose of the study is original but the study needs to be improved in the methodology.

Author Response

Reviewer 3:

We would like to thank the reviewer for finding our manuscript interesting. We devoted considerable attention to the reviewer comments and revised the manuscript accordingly.

  • Title

Title is appropriate because it is completely informative about the contents of the paper.  

Answer: Thank you

  • Abstract

The abstract respects the rules of the journal. The background and the aim are interesting. In the design is present the type of study. Setting, population and methods need to be better explained. The clinical Impact is present but need to be better explained.

Answer: We would like to thank the reviewer for this comment. The abstract was re-written.

Abstract: The study aimed to evaluate chronic low back pain (CLBP) patients' outcome measures and perceptions after being treated with the Enhanced Transtheoretical Model Intervention (ETMI). In a process evaluation mix method study, 30 CLBP patients completed electronically self-reported measures (function, pain, and fear-avoidance belief) before and after an ETMI treatment. Subsequently, each of the patients participated in a one-on-one semi-structured interview, which were audio-recorded, transcribed, coded, and analyzed thematically. Quantitative analysis showed significant improvements in function [P<0.001], pain [P<0.001], and fear-avoidance beliefs [P<0.001] after receiving ETMI treatment. With a large effect size [Cohen's d=1.234]. Moreover, the average number of physiotherapy sessions was 2.6±0.6 for the ETMI intervention while the annual average number in Maccabi is estimated at 4.1±1.5. Three main themes emerged from the thematic analysis: 1) Communication between the patient and the practitioner; 2) Psycho-Social treatment elements and 3) ETMI as a long-term solution for CLBP. The findings of the current study highlight patients' perceived need for an open and sincere dialog and for receiving reassurance and encouragement about their LBP. Of importance, they had no problem with the fact that they did not receive passive treatment. Accordingly, together with significant improvement in post-treatment outcome measures, patients perceived the ETMI method as a practical tool for self-managing their back problems in the long term.

Text

  • The introduction of the study does clearly sum up the background of the study.

Answer: Thank you

  • The authors provide a rationale for performing the study based on a review of the medical literature, but they can improve it.

Answer: the introduction was improved in accordance to the reviewer comment (page 2, lines 82-89).

Recent studies address issues around patients' and physiotherapists (PT's) perceptions and expectations of current physiotherapy and of the ETMI method [9,10]. While patient's primary purpose is to reduce their pain, they expect to receive also passive treatment, and information and knowledge regarding their back problem, [9] PT's indicated that there are barriers to overcome, such as PTs communication and education, individual's reluctance to change their daily routine, interprofessional collaboration and health care complexity [10].

  • Furthermore, they do not define well terms used in the remainder of the manuscript. (Please add a list of abbreviations before References section to your manuscript).

Answer: a list of abbreviations before References was added (Page 14, Line 482).

Abbreviations

BMI: Body Mass Index CI: Confidence Interval; CLBP: Chronic Low Back Pain; ETMI: Enhanced Transtheoretical Model Intervention; FABQ: Fear Avoidance Belief Questionnaire; FOTO: Focus on Therapeutic Outcomes; GP: General Practitioner; LBP: Low Back Pain; LCAT: Lumbar Computerized Adaptive Test; MCID: minimal clinical important difference; MHS: Maccabi Healthcare Services; NSAID: Non-Steroidal Anti-Inflammatory Drugs; NPRS: Numeric Pain Rate Scale; PTs: Physiotherapists; SD: standard deviation.  

  • The hypothesis is defined.

Answer: Thank you

  • The methods are not clear about the statistical method.  

First, the authors can add a subtitle of sample size estimation right behind study population; based on what did you calculate this sample size?

Answer: A sample size estimation was added in the statistical analysis section. (Page 6, Line 210-214)

2.5.1. Sample size

The sample size was calculated with G*Power 3.1.9.4 for a paird t-test to detect the difference between two dependent means. The input parameters were as follows: for a single-tailed test, assuming a medium effect size of 0.5, α= 0.05 and power= 0.8, the total sample size recommended was 27 participants.

  • The methodology needs to be improved a bit.

Answer: most of the method section was re-written.

  • About the patients: how many patients are recruited? Because it is not clear. This is a crucial information.

Answer: We added the exact number of patients that were recruited for this study.

30 LBP patients admitted to physical therapy consultation (Page 4, Line 141)

  • About the centers: Which ones are they?

Answer: in accordance with the reviewer comment we added the physiotherapy clinical center -Ramat-Gan. (Page 4, Line 142)

  • The results are reported clearly and concisely.

Answer: Thank you

  • In the discussion, it is important to emphasize the importance of this method explaining the specific aspects in the daily practice.

Answer: We would like to thank the reviewer for this important comment. We have emphasized to importance of the use of mixed method study in the first paragraph in the discussion section. (Page 12, line 386)

This study focused on CLBP patients’ perceptions and outcome measures following physiotherapy treatment using the ETMI method. By using mix-method research, this study obtained a more comprehensive view of patients' treatment perceptions, relationships with clinicians, and lived experience of their problems following physiotherapy treatment given through the ETMI method [35,36].

  • Limitation:

To be explained better.

Answer: the limitation section was re-written in accordance to reviewer suggestion. (Page 13, Line 444)

The interviews were conducted by the same PT who provided the treatment according to the ETMI method. This may entail a risk for social desirability and interviewer biases [33]. However, patients were discharged, no dropout was recorded and they did not return to another treatment session after 3 months of follow up.

Furthermore, to reduce the risk, all the quantitative data was collected privately without the presence or any assistance from the interviewer.

  • Tables

They sum up the study concisely and clearly.

Answer: Thank you

  • Figures

They sum up the study concisely and clearly.

Answer: Thank you

  • General comments

The purpose of the study is original, but the study needs to be improved in the methodology.

Answer: We would like to thank the reviewer for finding our manuscript interesting. We hope we have been improved the overall study methodology and revised the manuscript accordingly to the reviewer important comments.

Round 2

Reviewer 1 Report

The authors have improved the quality of their work significantly in the revised version.

Reviewer 3 Report

The authors met all my requests.